# Enhanced Bioactive Properties of Halloysite Nanotubes via Polydopamine Coating

**DOI:** 10.3390/polym14204346

**Published:** 2022-10-15

**Authors:** Mehtap Sahiner, Sahin Demirci, Nurettin Sahiner

**Affiliations:** 1Department of Bioengineering, Faculty of Engineering, Canakkale Onsekiz Mart University Terzioglu Campus, Canakkale 17100, Turkey; 2Department of Chemistry & Nanoscience and Technology Research and Application Center, Canakkale Onsekiz Mart University Terzioglu Campus, Canakkale 17100, Turkey; 3Department of Ophthalmology, Morsani College of Medicine, University of South Florida, 12901 Bruce B. Downs Blv., MDC 21, Tampa, FL 33612, USA

**Keywords:** halloysite nanotube composite, polydopamine coating, antioxidant natural clay, blood compatible, surface modified HNT with PDOPA

## Abstract

Halloysite nanotubes (HNT) were coated five times with dopamine (DOPA) in a tris buffer medium at pH 8.5 to acquire polydopamine-coated HNTs (PDOPA@HNT), e.g., PDOPA1@HNT, PDOPA3@HNT, and PDOPA5@HNT. Upon coating HNT with PDOPA, the surface area, pore volume, and pore size were decreased depending on the number of coatings. While the surface area of HNT was 57.9 m^2^/g, by increasing the number of coatings from 1 to 5, it was measured as 55.9, 53.4, 53.3, 47.4, and 46.4 m^2^/g, respectively. The isoelectric point (IEP) for HNTs was determined as 4.68, whereas these values are estimated as 2.31 for PDOPA1@HNTs, 3.49 for PDOPA3@HNT, and 3.55 for PDOPA5@HNT. Three different antioxidant studies were conducted for HNT and PDOPA@HNT, and the total phenol (TPC) value of HNT was found to be 150.5 ± 45.9 µmol gallic acid (GA) equivalent. The TPC values for PDOPA1@HNT, PDOPA3@HNT and PDOPA5@HNT coatings were found to be 405.5 ± 25.0, 750.0 ± 69.9, and 1348.3 ± 371.7 µmol GA equivalents, respectively. The Fe(II) chelation capacity of HNT was found to be 20.5% ± 1.2%, while the PDOPA1@HNT, PDOPA3@HNT and PDOPA5@HNT values were found to be 49.9 ± 6.5, 36.6 ± 12.7 and 25.4 ± 1.2%, respectively. HNT and PDOPA@HNTs inhibited the α-glucosidase (AG) enzyme to greater extents than acetylcholinesterase (AChE). As a result, the DOPA modification of HNTs was rendered to provide additional characteristics, e.g., antioxidant properties and higher AChE and AG enzymes inhibition capabilities. Therefore, PDOPA@HNTs have great potential as biomaterials.

## 1. Introduction

Halloysite nanotubes (HNTs) are natural clays with tubular nanostructures that have outstanding potential in many fields such as biomedicine, cosmetics, and the environment. An HNT consists of a multilayered structure composed of tetrahedral SiO_4_ layers and octahedral AlO_6_ layers arranged in 1:1 ratio [1,2]. Scientific interest in HNTs has continuously increased as they are modifiable, inherently non-toxic, natural, biocompatible [3,4], and in most cases are inexpensive and reasonable in device application [5,6], as they are abundant in nature. Neat HNTs are low cost and have high availability, and pore sizes can range from mesopore (2–50 nm) to macropore (>50 nm) [7]. They possess superior tubular structures [8,9,10] compared to other synthetic nanotubes, e.g., carbon nanotubes, and they may also afford some advantages, as they are easily chemically modifiable. HNTs are used in cosmetics [11], nanomedicine [12], tissue engineering [13], electrochemical sensors [14], catalytic and environmental applications [15] due to their negative charged outer and positively charged inner tubular properties with varying functionality [16]. Recently, there have been an increased number of studies in the use of HNTs in drug delivery systems by forming nanohybrid composites with organic compounds such as chitosan, alginate, cellulose and cyclodextrin [17,18,19]. In a study, polyethyleneimine (PEI) was used to modify HNTs, and upon the modification of HNTs, the Van der Walls force is reduced, and the cation exchangeability is increased, which resulted in the better dispersion of m-HNTs in Acrylonitrile–Butadiene–Styrene (ABS) copolymer blends [20]. Moreover, it was reported that octylphosphonic acid (OPA) can be used to functionalize HNTs, and the effects of surface charge on the localization and reversibility of the OPA were investigated [21]. Furthermore, natural cellulose and HNTs were used to fabricate HNT-based supercapacitors [22]. In addition, HNTs were used as support material in the preparation of a bimetallic palladium catalyst for the oxidation of carbon monoxide [23].

Dopamine (DOPA) is a monoamine and has a catechol and amine group [24,25]. Proteins of mussels contain DOPA [26], and it is an important neurotransmitter in the human body providing electrical signal transmission between nerve cells [27,28]. Therefore, the deficiency of DOPA can cause many nerve and mental-related diseases [29]; hence, some studies reported on the use of DOPA as a drug [30]. Interestingly, DOPA is frequently used as a coating material due to its adhesive properties, good biocompatibility, and high reactive characteristics [31] for monolithic column [32], cotton fabric, and titanium implants [33]. As there is no steric hindrance by dopamine or its oxidized oligomers when they attach to narrow surfaces [34], DOPA can readily bind to surfaces in a single step [35]. DOPA and the similar chemical structure of norepinephrine can render new functionality to the materials by covering almost any surface [36]. It was also demonstrated that DOPA promoted osteoblast growth in implant applications [37]. Likewise, DOPA was used to coat a poly(ether sulfone) UF membrane to inhibit antifouling [38]. It was also reported that to enhance hydrophilicity and suspension stability, DOPA can be polymerized on the activated carbon electrodes [26].

Acetylcholinesterase (AChE) is an enzyme involved in brain functions, and it is converted to acetic acid and choline by the breakdown of the acetylcholine substrate [39]. In some aging-related diseases, this enzyme functions too much, rendering adverse effect. The inhibition of this enzyme is thought to contribute to Alzheimer’s disease [40]. Alpha-glucosidase (AG) is an enzyme that cleaves the 1,6 glycoside bonds of disaccharides. Disaccharides cannot be absorbed from the gastrointestinal tract. However, the enzymes such as amylase and AG that break down sugar, e.g., polysaccharides and disaccharides, can be broken down into monosaccharide units by these enzymes; then, the glucose is absorbed by the intestines. There are multiple treatment methods for the treatment of hyperglycemia. One of these methods is inhibition of the AG enzyme. Some flavonoids are known to Inhibit the AG enzyme [41,42]. The AG effect of biomolecules consumed by the body is very important. In diabetes, especially in type II, there is an abnormal increase in the level of glucose in the blood, and related cardiovascular diseases such as hypertension occur [43]. It is important to treat such diseases by inhibition of the AG enzyme [44]. The inhibition of AG enzymes in controlling high blood sugar after eating is one of the methods used for the treatment of type II diabetes [45].

In this study, halloysite nanotubes (HNTs) were coated with poly(DOPA) (PDOPA) one, three, and five times sequentially as PDOPA1@HNT, PDOPA3@HNT and PDOPA5@HNT and then characterized. Fourier transform infrared radiation (FT-IR) spectroscopy, thermal gravimetric analysis (TGA), surface area and porosity instruments were used in the characterization studies. The antioxidant capabilities of HNT and DOPA coated HNTs were tested via the total phenol (TPC), total flavonoid content (TFC) and Fe(II) chelating studies. The zeta potential values of HNT and PDOPA@HNT-coated materials at different pH values were investigated to assess the isoelectric points (IEPs). The enzymes such as AG and AChE as well as the enzymes inhibition capabilities of HNT and PDOPA@HNTs were investigated. Various surface modification and coatings techniques such as grafting, etching surface functionalization via chemical treatment, irradiation, and so on are commonly used to obtain surface-modified material with new properties. Specifically, to acquire phenolic coatings of the surfaces, catechol-based monomers are commonly utilized to obtain materials with antioxidant properties. To determine the antioxidant properties of the materials, tests such as total phenolic content (TPC), ABTS+ scavenging assays, total flavonoid content (TFC), and iron (II) ion-chelating capacity methods are normally used [41]. In this study, in addition to the total phenol content (TPC), total flavonoid content (TFC), and iron (II) ion-chelating capacity of PDOPA@HNTs, the AG and AChE enzyme inhibition capabilities of PDODA@HNTs were determined to assess the potentials of PDODA@HNTs materials as an alternative to the traditional drugs used in the treatment of diseases such as diabetes and Alzheimer’s.

## 2. Materials and Methods

### 2.1. Materials 

Halloysite nanotubes (HNT, Al_2_Si_2_O_5_(OH)_4_·2H_2_O, Aldrich, St. Louis, Mo, USA) and dopamine hydrochloride (DOPA, 98%, Sigma) were used as received. Tris (99%, Sigma Aldrich, St. Louis, Mo, USA) was used to prepare a buffer solution for polymerizing DOPA during HNT coating. Sodium nitrite (NaNO_2,_ extra pure, Merck, Darmstadt, Germany) and aluminum chloride hexahydrate (AlCl_3_·6H_2_O, 99%, Alfa Aesar, Haverhill, MA, USA) were used for the total flavonoid test. Folin–Ciocalteu phenol reagent (FC) (Sigma-Aldrich, St. Louis, Mo, USA) was used for total phenol test. Gallic acid (GA) (97.5–102.5%, Aldrich) was used as a reference phenol agent. Iron(II) sulfate heptahydrate (FeSO_4_7H_2_O, 99.5%, Merck, Darmstadt, Germany), and 5,6-diphenyl-3-(2-pyridyl)-1,2,4-triazine-4,4-disulfonic acid disodium salt hydrate (Ferrosine, 99%, Alfa Aesar, Haverhill, MA, USA) were used for Fe(II) chelation assay. α-Glucosidase from Saccharomyces cerevisiae (100 unit/mg, Sigma-Aldrich, St. Louis, Mo, USA), 4-nitrophenyl-α-D-glucopyranose (4-NPG, 99%, Acros Organics, Geel, Belgium), and potassium phosphate monobasic (98–100.5%, Sigma Aldrich) were used to study an AG enzyme inhibition test. Acetylcholinesterase (AChE, Type VI-S, 200–1000 units/mg protein, Sigma Aldrich) from electric eel Electrophorus electricus, bovine serum albumin (BSA, Fisher Scientific, Hampton, NH, USA), 5,5-dithiobis-2-nitrobenzoic acid (>98%, TCI America, Portland, OR, USA), acetylthiocholine iodide (>98%, TCI America), Trizma^®^ hydrochloride (Sigma, St. Louis, MO, USA), magnesium chloride hexahydrate ((Cl_2_Mg·6H_2_O, Fisher Scientific, Hampton, NH, USA), and sodium chloride (NaCl, crystalline/certified ACS, Fisher Scientific, Hampton, NH, USA) were used for AChE enzyme studies. 

### 2.2. Coating of HNTs with PDOPA

The coating of HNTs with PDOPA was accomplished by the self-polarization of DOPA in tris buffer according to the literature [46,47]. First, the HNTs were washed for 24 h with an ethanol/water (50:50, *v*/*v*) mixture to remove any unexpected residue that may have developed during the recovery processes. The HNTs were cleaned by mixing 10 g of HNT in 1 L of ethanol/water (50:50, *v*/*v*) mixture at room temperature and stirring at 1000× *g* rpm for 24 h. Next, the washed HNTs were precipitated by centrifugation of the ethanol/water mixture at 10,000× *g* rpm and freeze-dried at −86 °C and 0.011 mbar. Thereafter, 10 g of dried HNT was dispersed into 500 mL of freshly prepared DOPA solution at a concentration of 2 mg/mL in 10 mM Tris buffer at pH 8.5. Next, the self-polymerization reaction of DOPA was conducted at 1000× *g* rpm for six hours to coat the HNTs with PDOPA. Lastly, the PDOPA-coated HNTs (PDOPA1@HNT) were separated by 10 min centrifugation at 10,000× *g* rpm and washed twice with DI water. After removing 2.0 g of PDOPA-1st@HNT, the remaining 8.0 g of PDOPA1@HNT was recoated with PDOPA as before. In succession, five coatings were applied, and after each coating, 2.0 g of PDOPA@HNT was separated and washed. Each DOPA coating of HNTs was completed at room temperature using 2.0 mg/mL DOPA solution in 10 mM tris buffer medium under 1000× *g* rpm mixing rate for six hours. Prepared PDOPA2@HNT, PDOPA3@HNT, PDOPA4@HNT and PDOPA5@HNT samples were washed and dried. Finally, the PDOPA@HNT samples were stored in closed tubes for characterization and other uses.

### 2.3. Characterization of PDOPA@HNTs

The spectra for the functional groups in HNT and PDOPA@HNT were evaluated with Fourier transform infrared radiation (FT-IR, Nicolet iS10, Thermo, Boston, MA, USA) using the attenuated total reflection (ATR) technique in the range of 650–4000 cm^−1^ in 4 repetitions. The percentage of PDOPA coated on the HNTs was determined using a thermogravimetric analyzer (TGA, SII TG/DTA6300, Exstar, Seiko Ins. Corp, Tokyo, Japan.). Thermogravimetric analyses were carried out by heating about 4 mg of PDOPA@HNT sample from 100 to 500 °C with a temperature increase of 10 °C/min in the presence of 200 mL/min nitrogen gas flow. X-ray diffraction pattern (X-RD) analysis of HNTs and PDOPA@HNTs was carried out using a Cu Kα X-ray source (40 kV, 40 mA) at 1.5418 Å with a Bruker D8 in the scanning range of 10–60° (2θ) Advance using a diffractometer. The surface area variation and porosity properties of HNTs and PDOPA@HNTs were examined by N_2_ adsorption/desorption measurements (TriStar II, Micromeritics, Norcross, GA, USA). While the surface area of HNT-based materials was determined by the BET method, the pore size and pore volumes were calculated according to the BJH method. HNT-based materials were degassed with N_2_ gas for 12 h before the analysis. The N_2_ gas adsorption/desorption curves were obtained in liquid nitrogen. The zeta potential measurements of the HNT and PDOPA@HNT sample were performed using the Zeta Potential Analyzer (Zeta-Pals, BIC) in 0.001 M KCl solutions suspended in DI water at a concentration of 1.0 mg/mL. In addition, the isoelectric points were calculated by measuring the surface charges of HNT and PDOPA@HNT structures at different pH values.

### 2.4. Antioxidant Properties of PDOPA@HNTs

The antioxidant activities of HNT-based materials were determined using the total phenol content (TPC), total flavonoid content (TFC) and iron (II) ion-chelating capacity methods.

The total amount of phenol content (TPC): TPC for HNT and PDOPA@HNTs was determined in accordance with the literature for 2.0 mg/mL aqueous solutions [41]. First, 0.1 mL solution from 2.0 mg/mL aqueous PDOPA@HNT suspension was mixed with 1.25 mL of 10% Folin–Ciocalteu (FC) reagent solution in DI water. Saturated sodium carbonate aqueous solution was added to this solution after 4 min and incubated at room temperature for 2 h in the dark. After incubation with gallic acid (GA) as a standard, the concentration of gallic acid corresponding to the measured absorbance for each sample was determined by UV-Vis spectroscopy at a wavelength of 760 nm.

Total flavonoid content (TFC): TFC values of HNTs and PDOPA@HNTs were tested with a method available in the literature [48]. For this goal, 0.05 mL of 2.0 mg/mL PDOPA@HNT aqueous solution suspended in water was taken and placed in 96 wells, 0.025 mL of 3% NaNO_2_ solution was added to it and left for five minutes. Then, 0.025 mL of 6% AlCl_3_ solution was added. Afterwards, 0.1 mL of 1 M NaOH solution was added to the mixture. After 15 min of incubation, the mixture was measured at 415 nm with the Themo Multiskan Go microplate reader. Calibration was performed according to GA, and the values for PDOPA@HNTs are given according to their GA equivalent.

Iron(II) chelating capacity (ICC): In accordance with the literature [49], 140 µL of PDOPA@HNT aqueous solutions prepared at 2.0 mg/mL concentrations were put into a 96-well plate. Then, 20 µL of 1 mM of Fe(II) aqueous solution was added to them, and measurements were made using a microplate reader (Thermo Multiskan Go, USA) at 562 nm. Then, 40 µL of 2.5 mM ferrozine aqueous solution was added, and the measurement was performed again. DI water was used for control, and the results were calculated according to the literature [45].

### 2.5. Inhibition of AChE and AG Enzymes by PDOPA@HNTs

AChE enzyme interactions were performed for HNT and PDOPA@HNT at 2.0 mg/mL concentration in 50 mM tris buffer in accordance with the literature with some modifications [50] The test started with HNT and PDOPA@HNT prepared in 140 µL tris buffer and placed in a 96-well plate. Then, 3 mM 5,5-dithiobis (2-nitrobenzoic acid) (DTNB) was added to 50 mM tris buffer containing 20 µL of 0.02 M magnesium chloride and 0.1 M NaOH. Then, 0.022 units/mL of AChE enzyme in 20 µL of 0.1% BSA was added. After 10 min, absorbance at 405 nm was measured, and 20 µL (7.5 mM acetylcholine iodide) was added to a 96-well plate as substrate. After another 20 min, another measurement was taken. Each measurement was made in triplicate. 

AG enzyme inhibition was examined in phosphate buffer at varying concentrations (0–2.0 mg/mL) in accordance with the literature [47]. Briefly, 70 µL of HNT and phosphate buffer solution of PDOPA@HNT were put into 96 wells, and 70 µL of 0.03 unit/mL enzyme solution was added on top. The mixed solution absorbance at 405 nm was measured in a Thermo Scientific Multiskan Go microplate reader, and 70 µL of substrate solution was added. After 20 min, the measurement was taken again, and the inhibition value was calculated as percentage based on the phosphate buffer and given as in the literature [47].

The statistical analysis was performed according to *t* test for HNT and PDOPA@HNT at 2.0 mg/mL concentrations for AChE and AG enzyme inhibition tests, and the Origin Pro 2021 program was used in the statistical analysis.

### 2.6. Blood Compatibility of HNT with Multiple PDOPAs

The blood compatibility of the prepared PDOPA@HNTs was determined by hemolysis and blood coagulation tests in accordance with the procedures available in the literature [41]. Blood compatibility studies were carried out with the approval of the ethics committee (no. KAEK-2011-KAEK-27/2022-2200063689), which was approved by the Human Research Ethics Committee of Çanakkale Onsekiz Mart University.

## 3. Results and Discussion

### Coating and Characterization of HNT with Multiple PDOPA Layers

An alkaline medium was widely used for the self-polymerization of DOPA [51,52]. Previously, it was observed that PDOPA began to precipitate after 8 h of self-polymerization reaction time, and this time period was accepted as a control study [47]. Hence, a 6 h reaction time was adopted as the reaction time to coat HNTs with PDOPA in this study. The coating of HNTs with PDOPA was carried out in tris buffer, pH 8.5 at room temperature. A schematic representation of the coating of HNTs with PDOPA is illustrated in Figure 1a.

In brief, HNTs were placed in a freshly prepared DOPA solution in 10 mM Tris buffer pH 8.5 at a concentration of 2.0 mg/mL. After stirring the mixture at 1000× *g* rpm for six hours, the resulting PDOPA@HNTs were centrifuged at 10,000× *g* rpm. This coating process was repeated five times consecutively. The PDOPA-coated HNTs were denoted as PDOPA1@HNT, PDOPA2@HNT, PDOPA3@HNT, PDOPA4@HNT and PDOPA5@HNT, respectively, the numbers showing the coating number. The photographs of HNT and PDOPA@HNTs are given in Figure 1b. As can be clearly seen, the white color of the HNT turned into black as the number of PDOPA coatings was increased on the HNT. This was associated with an increase in the amount of PDOPA coatings/layers on the HNT. After applying PDOPA coatings on the HNTs, the composite structures—e.g., PDOPA1@HNT suspended in water, 0.1 M HCl, 1 M HCl, 0.1 M NaOH, and 1 M NaOH, after sonication for 2 min—no elution of PDOPA was observed, as the PDOPA layers were tightly bound to the HNTs (data are not shown). This observation is in agreement with the literature [53]. 

The FT-IR spectra of HNT and PDOPA1@HNT, PDOPA3@HNT and PDOPA5@HNT were also compared, and the corresponding spectra are presented in Figure 2a to confirm the success of PDOPA coating of HNTs. In the FT-IR spectra of the HNTs, the peaks around 3620 and 3690 cm^−1^ were assigned to the stretching vibrations of the inner and outer surface hydroxyl groups of all HNTs, respectively. The peaks observed at 921 and 1003 cm^−1^ are also characteristic for all HNTs and are attributed to the Al-OH and Si-O-Si moieties, respectively [54]. In the FT-IR spectra of PDOPA@HNT structures, it was clearly seen that as the number of PDOPA coatings increased to one, three, and five, there was a decrease in the intensity of the peaks at 921 and 1003 cm^−1^, which was attributed to Al-OH and Si-O-Si vibrations, respectively. In addition, in the FT-IR spectra given in Figure 2a, the inset for the peaks in the range of 1700–1200 cm^−1^ is shown; here, the peaks seen at 1605 and 1514 cm^−1^ on the HNT increase with DOPA coatings because of N-H and indole ring stretching [55]. In addition, the peak at 1283 cm^−1^ whose intensity also increases with the increase in the number of coatings shows the C-O stretching vibrations extending from the phenolic groups in the PDOPA structure [55].

In addition, TGA thermograms of neat HNT and PDOPA@HNTs were also compared to confirm the increased amount of PDOPA on HNTs with multiple coatings, and the results are given in Figure 2b. Neat HNTs started to decompose at about 400 °C and lost 13.1% of their mass at 570 °C, while the total mass loss was 13.4% at 1000 °C. On the other hand, PDOPA-coated HNTs started to decompose at about 300 °C, and the mass losses were observed at 570 °C with 16.1%, 16.3%, 18.6%, 20.6% and 24.3% for PDOPA1@HNT, PDOPA2@HNT, PDOPA3@HNT, PDOPA4@HNT and PDOPA5@HNT, respectively. However, the total cumulative mass losses at 1000 ℃ were 16.5%, 17.5%, 19.0% 21.1% and 26.1% for PDOPA1@HNT, PDOPA2@HNT, PDOPA3@HNT, PDOPA4@HNT and PDOPA5@HNT, respectively. Here, the total mass loss was observed as 13.4% for HNT without coating at 1000 °C, while the total mass loss values observed at 1000 °C after coating with PDOPA increased, and the differences are measured by the percentage of the PDOPA amount in the coatings. Accordingly, after the first coating of HNT with PDOPA, 3.1% PDOPA was coated, and the amount of PDOPA coated on the HNT increased with successive coating processes; the amount of HNT coated after the 5th coating increased to 12.7%.

The effect of multiple PDOPA coatings of HNTs on the surface area and porosity properties of HNTs was also investigated, and the corresponding N_2_ adsorption/desorption isotherms were examined for bare HNT, PDOPA1@HNT, PDOPA2@HNT, PDOPA3@HNT, PDOPA4@HNT and PDOPA5@, respectively. The calculated surface area, pore volume and pore size values for HNT, PDOPA1@HNT, PDOPA2@HNT, PDOPA3@HNT, PDOPA4@HNT and PDOPA5@HNT are summarized in Table 1.

The surface area of neat HNT was measured as 57.9 m^2^/g and slowly decreased with the increase in the PDOPA coating number. Accordingly, the surface area of HNT was calculated as 55.9, 53.4, 53.3, 47.4 and 46.4 m^2^/g from the 1st to 5th coating layers, respectively. The decrease in the pore volume and pore size values of the multiple PDOPA-coated HNTs indicates that the pores of the HNTs were filled with PDOPA coatings processes. The decrease in the surface area of HNTs with the increase in the number of PDOPA coatings may be evidence that the HNT pores were filled by the PDOPA.

SEM images of HNT, PDOPA1@HNT and PDOPA5@HNT particles are in Figure 3a. It is apparent that the thickness of neat HNT in comparison to PDOPA5@HNT is increased; therefore, the coating of HNTs with DOPA was successful, and the thickness of the HNTs increased as number of coatings increased.

In Figure 3b, the zeta potentials of HNT and PDOPA@HNT at different pH values are given. IEPs were calculated from the pH values that have zero zeta potential value in the graph. While the IEP point of HNT was at pH 4.68, it was determined as 2.31 for PDOPA1@HNT, 3.49 for PDOPA3@HNT, and 3.55 for PDOPA5@HNT. In addition, the pH values of PDOPA@HNTs were measured in 0.001 M KCl medium, and the pH of the neat HNT containing solution was measured as 7.1, whereas the pH values of 6.64, 6.35 and 6.09 were determined for PDOPA1@HNT, PDOPA3@HNT, and PDOPA5@HNT, respectively. The pH values shift toward the acidic direction as the number of coatings increases, indicating the presence of some protonated amine groups. As the number of coatings was increased, the colloidal stability of the composites was decreased as each layer of the polymer coating of DOPA generated bigger particles, which can easily precipitate under gravity. In addition, the PDOPA coatings of HNTs can induce intra particle polymer–polymer interactions between the PDOPA-coated HNTs.

In this study, the XRD analysis of HNT, PDOPA3@HNT and PDOPA5@HNT coatings was performed, and the results are given in Figure 4. The X-ray diffraction (XRD) pattern for HNT revealed a peak at 11.91° (001), a peak at 20.17° (100) and a peak at 24.8° (002) as indicative of the halloysite structure. The peak at °2θ reflects the characteristic crystal structure of halloysite-10 Å. Upon PDOPA coatings, the crystal structure of the HNTs was not affected, and the tubular nanostructure was preserved for PDOPA3@HNT and PDOPA5@HNT coatings [56]. The intensity of diffractions was increased by the addition of PODA layers onto the existing crystal structure, as shown in Figure 4.

Three different antioxidant tests were performed for PDOPA@HNTs. The TPC value was determined for aqueous solutions of PDOPA@HNTs at a concentration of 2.0 mg/mL. As seen in Figure 5a, the TPC value of HNT was determined to be 150.5 ± 45.9 µmol GA equivalent (eq). TPC values for PDOPA1@HNT, PDOPA3@HNT and PDOPA5@HNT coatings were found as 405.5 ± 25.0, 750.0 ± 69.9 and 1348.3 ± 371.7 µmol GA eq, respectively. As the number of coatings was increased, the TPC values were also increased almost linearly.

The TFC values for PDOPA@HNT coatings are given in Figure 5b. The highest value was determined as 187.3 ± 10.8 µg/mL GA equivalent in PDOPA3@HNTs. In PDOPA5@HNTs, it was found as 150.3 ± 9.4 µg/mL GA eq., whereas the neat HNT had a TFC value of 25.6 ± 20.3 µg/mL GA eq. The reason why PDOPA3@HNT had a higher value than PDOPA5@HNT may be due to the aggregation of PDOPA layers as the number of coatings increased.

On the other hand, the percentage of Fe(II) chelation is given in Figure 5c, and it was observed that the chelating capacity decreased with the increase in the number of DOPA coatings of HNTs. This may be due to the reduced number of functional groups that can contact Fe(II) ions due to DOPA aggregation, or it could be due to the decrease in the number of functional groups on DOPA that can interact with Fe(II), leading to reduced chelating capacity.

Figure 6a shows the AChE enzyme interactions of PDOPA@HNTs at 2.0 mg/mL concentration for 0.022 units/mL of AChE enzyme. As the number of coatings increased, the enzyme inhibition capability also increased, and the highest inhibition was achieved with PDOPA5@HNT at a value of 34.3 ± 0.2%. On the other hand, HNT had almost non-existent enzyme inhibition ability with the value of 2.9 ± 0.9% enzyme inhibition at the same concentration. According to the *t* test, HNT at 2.0 mg/mL concentration was statistically different from PDOPA1@HNT, PDOPA3@HNT, and PDOPA5@HNT in AChE enzyme inhibition studies. 

It is known in the literature that HNTs inhibit the AG enzyme [57]. Here, the AG enzyme inhibition ability of PDOPA@HNT was also studied, and we compared the neat HNTs with PDOPA@HNTs at the same concentration (2.0 mg/mL). As shown in Figure 6b, we observed that both HNT and PDOPA@HNTs composites clays inhibited the AG enzyme close to 100% at 2.0 mg/mL concentrations. Statistical analysis according to t-test for HNT and PDOPA@HNTs at 2.0 mg/mL concentrations in AG inhibition studies was performed. According to Figure 6b, HNT and PDOPA1@HNT at 2.0 mg/mL concentration were not statistically different. However, for the same concentrations, HNT is statistically different from PDOPA3@HNT and PDOPA5@HNT (* *p* < 0.05). 

Different concentrations of neat HNT and PDOPA@HNT coatings were studied for AG enzyme inhibition. As demonstrated in Figure 6c, the AG enzyme inhibition abilities of neat HNTs and PDOPA@HNTs in the concentration range of 0.125–1 mg/mL are almost in a linear dependence with their concentrations. For example, HNT inhibited AG enzyme 40.8 ± 3.5% at 0.5 mg/mL concentration, whereas at the same concentration (0.5 mg/mL), PDOPA1@HNT, PDOPA3@HNT and PDOPA5@HNT were found to inhibit as 56.2 ± 2.2%, 18.5 ± 4.3% and 30.3 ± 7.6%, respectively.

The blood compatibility of neat HNT and PDOPA@HNTs was investigated using 2.0 mg/mL particle concentration by hemolysis and blood coagulation tests. As shown in Figure 7a, the hemolysis index was less than 5% for all samples. The highest value was 0.8 + 0.5, while the others were even lower; therefore, neat HNTs and PDOPA@HNTs can be accepted as blood-compatible materials with low hemolysis indexes. The blood coagulating test results for HNT and PDOPA@HNTs are also given in Figure 7b.

The blood coagulating index percentage values of neat HNTs and PDOPA HNTS was found to vary between 95.3% and 97.2%. The highest value was 97.2 ± 2.1% for PDOPA@3HNT, while the neat HNT had a value of 95.3 ± 4.0%. Therefore, it is apparent that neat HNT and PDOPA@HNTs do not induce blood clotting and are suitable for blood contacting applications.

## 4. Conclusions

It was shown that HNT clays can be successfully coated with PDOPA multiple times in Tris-buffer solution to render addition properties. Coated HNTs possessed additional new properties such as antioxidant properties, which is an important feature of some biomaterials to scavenge radicals that are responsible for many diseases. As the HNTs were coated with PDOPA, the pore size, pore volume and surface area were decreased, while their enzyme inhibition capacity for the AChE enzyme of PDOPA1@HNT, PDOPA3@HNT and PDOPA5@HNT was enhanced significantly. In contrast, the AG enzyme inhibition properties improved slightly at 2.0 mg/mL concentration. It was further demonstrated that the PDOPA coating did not influence the blood compatibility of DOPA-coated HNTs, as the hemolysis percentage and blood coagulation index percentage values were found in the acceptable ranges which is 2.5% for hemolysis percentage and ≥95% for blood coagulation index percentage. Therefore, coating HNTs with PDOPA for up to five layers affords great potential in the biomedical field with antioxidant capability, different enzyme inhibition ability and blood compatibility. In the studies of AChE inhibition, which is very important in Alzheimer’s disease, it was found that the effectiveness of AChE inhibition was increased as the number of PDOPA coatings on HNT increased. Likewise, as the number of PDOPA coatings increased, the antioxidant property of PDOPA@HNTs increased. In future studies, the loading of various Alzheimer’s drugs into PDOPA@HNTs composites is planned to be investigated, as PDOPA@HNTs composites have promising potential in the treatment of diseases, e.g., age-related Alzheimer’s, and these materials can potentially be used for the release of the relevant active agent into brain cells in various animal models.

## Figures and Tables

**Figure 1 polymers-14-04346-f001:**
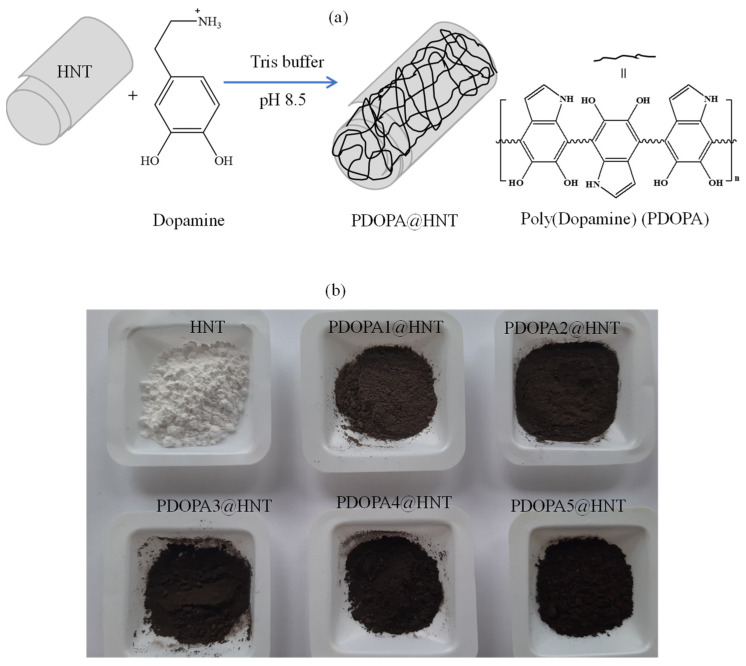
(**a**) Schematic representation of PDOPA coating of HNTs, and (**b**) digital camera photographs of PDOPA@HNT.

**Figure 2 polymers-14-04346-f002:**
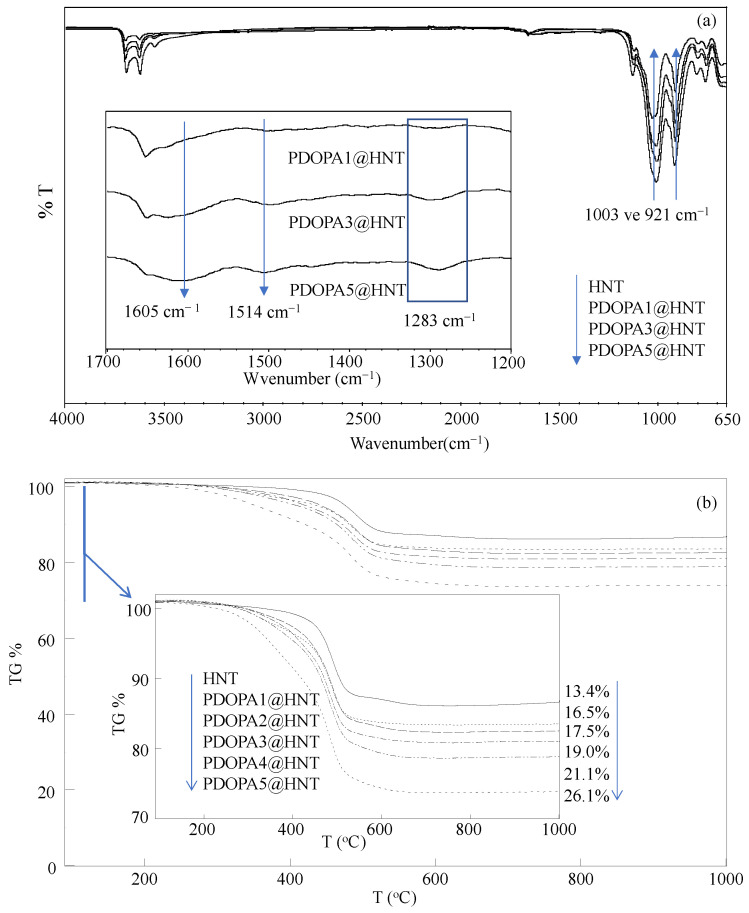
(**a**) FT-IR spectra and (**b**) TGA thermograms of HNT and PDOPA@HNT samples.

**Figure 3 polymers-14-04346-f003:**
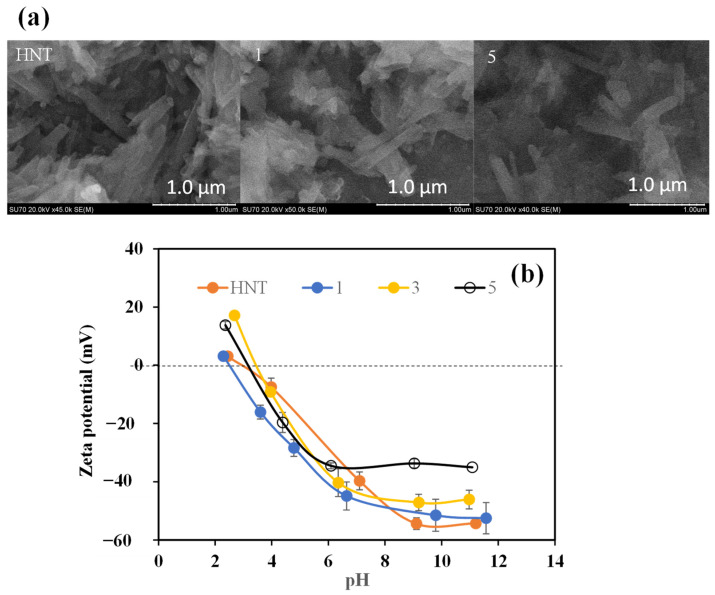
(**a**) SEM images of HNT, PDOPA1@HNT and PDOPA5@HNT clays, and (**b**) zeta potential values for HNT, PDOPA1@HNT, PDOPA3@HNT, PDOPA5@HNT at different pH values (scale bar 1.00 µm in the SEM images).

**Figure 4 polymers-14-04346-f004:**
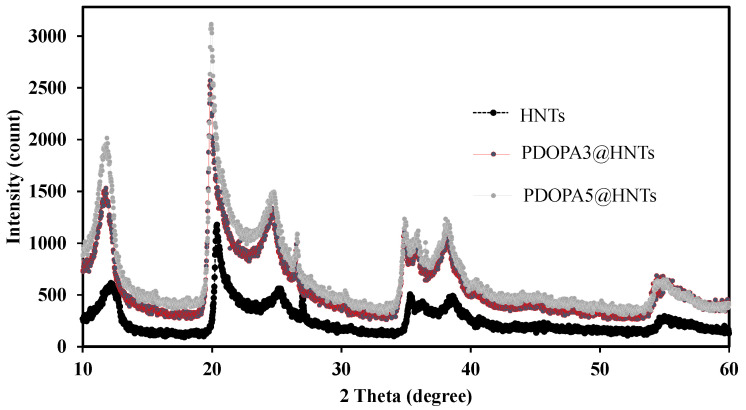
XRD patterns of HNT, PDOPA3@HNT, and PDOPA5@HNT particles.

**Figure 5 polymers-14-04346-f005:**
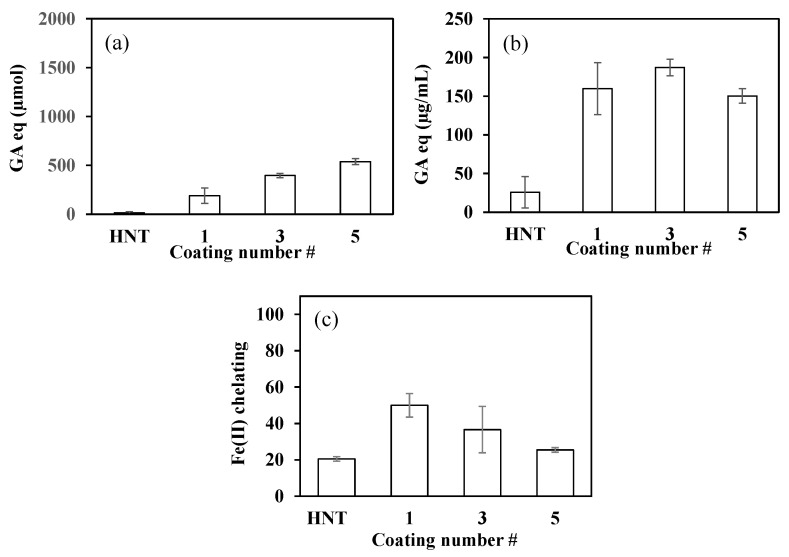
(**a**) TPC values, (**b**) TFC values, and (**c**) Fe(II) chelating capacities (%) of HNT and PDOPA@HNT particles.

**Figure 6 polymers-14-04346-f006:**
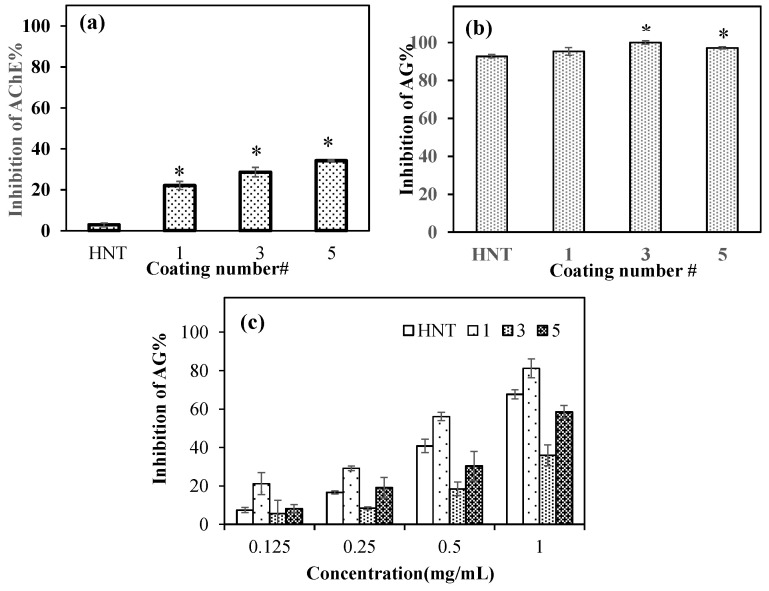
(**a**) AChE and (**b**) AG enzyme inhibitions by HNT, PDOPA1@HNT, PDOPA3@HNT and PDOPA5@HNT at 2.0 mg/mL concentration, a t-test was used to determine if PDOPA@HNT were significantly different from bare HNT (* *p* < 0.05 vs. control), and (**c**) AG enzyme inhibitions by HNT, PDOPA1@HNT, PDOPA3@HNT and PDOPA5@HNT at various concentrations (0.125, 0.25, 0.5, and 1.0 mg/mL).

**Figure 7 polymers-14-04346-f007:**
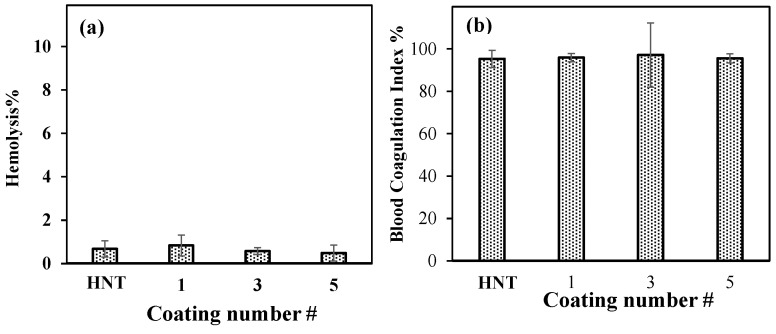
(**a**) Hemolysis percentage and (**b**) blood coagulation index percentage of neat HNT and PDOPA1@HNT, PDOPA3@HNT and PDOPA5@HNT at 2.0 mg/mL concentration.

**Table 1 polymers-14-04346-t001:** The effect of multiple PDOPA coatings on surface area, pore volume and pore size values of HNTs.

Materials	BET Surface Area (m^2^/g)	Pore Volume (cm^3^/g)	Pore Size (nm)
HNT	57.9	4.1	31.3
PDOPA1@HNT	55.9 ± 0.3	4.2	34.2
PDOPA2@HNT	53.4	4.5	31.5
PDOPA3@HNT	53.3	4.3	32.9
PDOPA4@HNT	47.4	4.1	24.6
PDOPA5@HNT	46.4	3.9	18.5

## Data Availability

The data presented in this study are available on request from the corresponding author.

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
