# Peer review of "Enhanced Bioactive Properties of Halloysite Nanotubes via Polydopamine Coating"

_polymers, 2022, doi:10.3390/polym14204346_

Round 1
Reviewer 1 Report
The manuscript "Enhanced Bioactive Properties of Halloysite Nanotubes via Pol-2 ydopamine Coating" is well written. However, it needs improvement:
1. Provide detailed methodology in the Materials & Method section.
2. In results and Discuss section, provide high-resolution SEM images for better understanding.
3. Provide the The state-of-the-art comparisons for the proposed work are missing in this paper. Then do a critical analysis of previous research. State explicitly the shortcomings of previous research. What is positive in previous research and what is negative. Based on that, you explicitly define the goal of the research and the scientific hypothesis.
4. The biggest shortcoming of the research is that there is no analysis of errors, analysis of sensitivity of results and analysis of uncertainty of results.
5. The Conclusion section should be rewritten. Highlight your scientific contribution. Highlight the benefits of your research. Define shortcomings and future research.
Author Response
Reviewer 1
Comments and Suggestions for Authors
The manuscript "Enhanced Bioactive Properties of Halloysite Nanotubes via Pol-2 ydopamine Coating" is well written. However, it needs improvement:
-We are grateful for the reviewer time and effort and nice comments about the manuscript and improved her/his suggestions to improve manuscript by making the following changes/improvements;
1.Provide detailed methodology in the Materials & Method section.
- The methodology part is detailed also in subtitle 2.2 “Coating of HNTs with PDOPA” and a similar coating study is cited as ref #46. Also, the following information is added on pp 3 of the revised manuscripts as
“Each DOPA coatings of HNTs was done at room temperature using 2.0 mg/mL DOPA solution in 10 mM tris buffer medium under 1000 rpm mixing rate for six hours.”
2.In results and Discuss section, provide high-resolution SEM images for better understanding.
- Done as suggested. The SEM images were changed, and the scale bars made more visible.
3.Provide the The state-of-the-art comparisons for the proposed work are missing in this paper. Then do a critical analysis of previous research. State explicitly the shortcomings of previous research. What is positive in previous research and what is negative. Based on that, you explicitly define the goal of the research and the scientific hypothesis.
-The required information is added at the end of the last paragraph of the introduction section as;
“Various surface modification and coatings techniques such as grafting, etching surface functionalization via chemical treatment, irradiation, and so on are commonly used to obtain surface-modified material with new properties. Specifically, to acquire phenolic coatings of the surfaces, catechol-based monomers are commonly utilized to obtain materials with antioxidant properties. To determine the antioxidant properties of the materials tests such as total phenolic content (TPC), ABTS+ scavenging assays, total flavonoid content (TFC), and iron (II) ion chelating capacity methods are normally used [41]. In this study, in addition to the total phenol content (TPC), total flavonoid content (TFC), and iron (II) ion chelating capacity of PDOPA@HNTs, the AG and AChE enzyme inhibition capabilities of PDODA@HNTs were determined to assess the potentials of PDODA@HNTs materials as an alternative to the traditional drugs used in the treatment of diseases such as diabetes and Alzheimer's.”
4.The biggest shortcoming of the research is that there is no analysis of errors, analysis of sensitivity of results and analysis of uncertainty of results.
- Standard deviations for all the tests are included in the relevant figures.
On pp 5 of the revised manuscript, the following information is given about statistical analyses as;
“The statistical analysis was performed according to t test for HNT and PDOPA@HNT at 2.0 mg/mL concentrations for AChE and AG enzyme inhibition tests, and Origin Pro 2021 program was used in the statistical analysis.”
On pp 10 of the revised manuscript, the following information is provided;
“According to the t test, HNT at 2.0 mg/mL concentration was statistically different from PDOPA1@HNT, PDOPA3@HNT, and PDOPA5@HNT in AChE enzyme inhibition studies.”
Also, on pp 11 of the revised manuscript, the following information is provided;
“Statistical analysis according to t test for HNT and PDOPA@HNTs at 2.0 mg/mL concentrations in AG inhibition studies were performed. According to Fig 6 (b), HNT and PDOPA1@HNT at 2.0 mg/mL concentration were not statistically different. However, for the same concentrations, HNT is statistically different from PDOPA3@HNT and PDOPA5@HNT. (*p<0.05).”
- The Conclusion section should be rewritten. Highlight your scientific contribution. Highlight the benefits of your research. Define shortcomings and future research.
-The Conclusion section was improved as suggested and the following information is provided;
“In the studies of AChE inhibition, which is very important in Alzheimer's disease, it was found that the effectiveness of AChE inhibition was increased as the number of PDOPA coatings on HNT increased. Likewise, as the number of PDOPA coatings increased, the antioxidant property of PDOPA@HNTs increased. In future studies, the loading of various Alzheimer's drugs into PDOPA@HNTs composites is planned to investigate as PDOPA@HNTs composites have promising potentials in the treatment of diseases e.g., age-related Alzheimer's, and these materials can potentially be used for the release of the relevant active agent into brain cells in various animal models.”

Reviewer 2 Report
The manuscript entitled, ‘Enhanced Bioactive Properties of Halloysite Nanotubes via Polydopamine Coating’ reported preparation of polydoapmine coated HNTs and their biological belongings. I am mentioning some loopholes of this work which should be accounted before publication;
1. There are lots of articles on bioactive coating based on catechol based monomers. What is the most significant novelty here?
2. Why the authors choose HNTs irrespective of other anisotropic nanofillers?
3. In figure 1a, the structure of dopamine is overlapped. It should be clarified.
4. How did the author ensure that the coating of PDA was tightly bound to the HNTs surfaces? Is there any cross-check?
5. Did the author check the dispersibility or colloidal stability of the systems? Make some arguments on it.
6. Several articles related to catecholic coating have been reported in the literature which could be useful for reference: DOI https://doi.org/10.1039/C6RA24153K; https://doi.org/10.1021/acs.accounts.8b00583; https://doi.org/10.1021/acs.langmuir.2c00278; https://doi.org/10.1039/C1NR10969C.
Author Response
Reviewer 2
Comments and Suggestions for Authors
The manuscript entitled, ‘Enhanced Bioactive Properties of Halloysite Nanotubes via Polydopamine Coating’ reported preparation of polydoapmine coated HNTs and their biological belongings. I am mentioning some loopholes of this work which should be accounted before publication;
-We thank the reviewer for takin time and suggesting some important points to improve the quality of the manuscript and we improve the manuscript taking into his/her suggestions accordingly.
- There are lots of articles on bioactive coating based on catechol based monomers. What is the most significant novelty here?
- The part that emphasize the significance of this work was added at the end of Introduction section as;
“Various surface modification and coatings techniques such as grafting, etching surface functionalization via chemical treatment, irradiation, and so on are commonly used to obtain surface-modified material with new properties. Specifically, to acquire phenolic coatings of the surfaces, catechol-based monomers are commonly utilized to obtain materials with antioxidant properties. To determine the antioxidant properties of the materials tests such as total phenolic content (TPC), ABTS+ scavenging assays, total flavonoid content (TFC), and iron (II) ion chelating capacity methods are normally used [41]. In this study, in addition to the total phenol content (TPC), total flavonoid content (TFC), and iron (II) ion chelating capacity of PDOPA@HNTs, the AG and AChE enzyme inhibition capabilities of PDODA@HNTs were determined to assess the potentials of PDODA@HNTs materials as an alternative to the traditional drugs used in the treatment of diseases such as diabetes and Alzheimer's.”
- Why the authors choose HNTs irrespective of other anisotropic nanofillers?
- In the literature, the potential of anisotropic nanofillers can be used in many fields, including environmental applications, biomedical applications, paint, and aerospace materials are being commonly investigated. When incorporating these nanofillers, a number of factors were considered, including their mechanical characteristics, electrical conductivity, optical transparency, and barrier properties (Bandyopadhyay et al. 2020) and so on depending on their application field. In this study, some degenerative diseases and inhibition of different enzymes for the prevention of these diseases e.g., Alzheimer’s, Parkinson etc. were examined. Therefore, HNT is chosen because it is a biocompatible, natural, and inexpensive material that can be easily obtained.
The following information is given on pp 1 of the revised manuscript as;
“Scientific interest in HNTs has continuously increased as they are modifiable, inherently non-toxic, natural, biocompatible [3,4], and in most cases are inexpensive and reasonable in device application [5,6] as they are abundant in nature.”
-Bandyopadhyay, A.; Dasgupta, P.; Basak, S. Anisotropic Nanofillers in TPE; 1st ed.; Springer Singapore, 2020; ISBN 9789811590856.
- In figure 1a, the structure of dopamine is overlapped. It should be clarified.
- As suggested, Figure 1a has been rearranged as suggested.
- How did the author ensure that the coating of PDA was tightly bound to the HNTs surfaces? Is there any cross-check?
- As the number of PDOPA coatings increases on the HNT, the colloidal stability decreases. Aggregation and sedimentation of PDOPA@HNTs were observed in various solutions. As can be seen in the digital camera images of PDOPA@HNTs composite given below in various solution e.g., water, 0.1 M HCl, 1 M HCl, 0.1 M NaOH, 1 M NaOH and sonication after 2 min, there is no elution was observed as PDOPA layers were tightly bound to HNT.
Figure 1. The digital camera images of PDOPA1@HNT composites suspended in various solvents and sonicated 2 min.
Also, as mentioned, the white color of the HNT turned into black as the number of PDOPA coatings was increased. This observation is also in agreement with the literature, and a reference #53 is cited and the following information on pp 6 of the revised manuscript is given as;
“After PDOPA coatings of HNTs, the composites structures e.g., PDOPA1@HNT sus-pended in water, 0.1 M HCl, 1 M HCl, 0.1 M NaOH, 1 M NaOH, and sonication for 2 min, and there was no elution of PDOPA was observed as PDOPA layers were tightly bound to HNTs (data is not shown). This observation is in agreement with the literature [53].”
- Did the author check the dispersibility or colloidal stability of the systems? Make some arguments on it.
It cannot be said that PDOPA@HNTs show colloidal stability. For this reason, vortex and/or sonication was performed before each use to obtain the suspension for antioxidant and enzyme studies of PDOPA@HNT in.
The following information is provided on pp 9 of the revised manuscript as:
“...As the number of coatings was increased, the colloidal stability of the composites was decreased as each layer of the polymer coating of DOPA generated bigger particles which can easily precipitate under gravity. Also, the PDOPA coatings of HNTs can induce intra particle polymer-polymer interactions between the PDOPA coated HNTs.”
- Several articles related to catecholic coating have been reported in the literature which could be useful for reference: DOI https://doi.org/10.1039/C6RA24153K; https://doi.org/10.1021/acs.accounts.8b00583; https://doi.org/10.1021/acs.langmuir.2c00278; https://doi.org/10.1039/C1NR10969C.
The mentioned references were included on pp 2 and 6 the revised manuscript as;
“...As there is no steric hindrance by dopamine or its oxidized oligomers when they attach to narrow surfaces [34], DOPA can readily bind to surfaces in a single step [35]. DOPA and the similar chemical structure of norepinephrine can render new functionality to the materials by covering almost any surface [36]...”
“...After PDOPA coatings of HNTs, the composites structures e.g., PDOPA1@HNT sus-pended in water, 0.1 M HCl, 1 M HCl, 0.1 M NaOH, 1 M NaOH, and sonication for 2 min, and there was no elution of PDOPA was observed as PDOPA layers were tightly bound to HNTs (data is not shown). This observation is in agreement with the literature [53]...”

Round 2
Reviewer 1 Report
Accepted
Reviewer 2 Report
This can be published in its present form.